# Modeling the effects of perisaccadic attention on gaze statistics during scene viewing

Lisa Schwetlick [1,2✉], Lars Oliver Martin Rothkegel[1], Hans Arne Trukenbrod[1] & Ralf Engbert [1,2,3]

How we perceive a visual scene depends critically on the selection of gaze positions. For this selection process, visual attention is known to play a key role in two ways. First, image-features attract visual attention, a fact that is captured well by time-independent fixation models. Second, millisecond-level attentional dynamics around the time of saccade drives our gaze from one position to the next. These two related research areas on attention are typically perceived as separate, both theoretically and experimentally. Here we link the two research areas by demonstrating that perisaccadic attentional dynamics improve predictions on scan path statistics. In a mathematical model, we integrated perisaccadic covert attention with dynamic scan path generation. Our model reproduces saccade amplitude distributions, angular statistics, intersaccadic turning angles, and their impact on fixation durations as well as inter-individual differences using Bayesian inference. Therefore, our result lend support to the relevance of perisaccadic attention to gaze statistics.

[1] Department of Psychology, University of Potsdam, 14469 Potsdam, Germany. [2] DFG Collaborative Research Center 1294, University of Potsdam, 14469 Potsdam, Germany. [3] Research Focus Cognitive Science, University of Potsdam, 14469 Potsdam, Germany. ✉email: Lisa.Schwetlick@uni-potsdam.de

Visual perception in humans is the result of complex signal processing of visual input in the brain. Information enters the eyes at a rate of about $10^8 - 10^9$ bit/s[1]. In order to handle this enormous amount of input, the visual system relies on foveation and selective attention[2]. These two mechanisms reduce the information available at any given point in time to enable the brain to efficiently process the relevant aspects of visual information. *Foveation* refers to the decrease of visual acuity from the region extending about 2° around the point of fixation (the fovea) to the periphery of the visual field. During natural viewing, regions of interest are sequentially moved into the high-resolution foveal area by saccadic eye movements[3,4]. Natural vision is therefore an active process, determined by sequential choices of fixation locations. The resulting scan path[5] is characterized by pronounced spatial correlations[6]. *Selective attention* is the second key bottleneck of visual processing with a rate of about 100 bit/s[7], prioritizing selected image regions at the cost of others. Under natural viewing conditions, fixation position and visual attention are closely linked and coincide at the same location most of the time during viewing[3].

Experimentally, however, the locus of visual attention and fixation position can diverge, a condition referred to as covert attention[8,9]. Research on saccade dynamics in highly controlled experimental setups indicates that attention, as measured by processing benefits, precedes the fixation to the next saccade target[10–12]. Current models of eye movements and visual attention are typically based on the plausible simplification of directly equating location of attention and fixation position[13–16]. Here we propose that perisaccadic covert attention shifts are an important factor in eye-movement guidance. The field of modeling eye-movement behavior has primarily focused on predicting where fixations are placed in an image[17–19]. The most advanced models are able to predict fixation density maps that closely resemble the empirical fixation densities they are based on[20]. The step from modeling static fixation densities to predicting scan paths reveals that bottom-up image information, while important, cannot comprehensively explain the fixation selection process. This is illustrated by the fact that even a model that comprises no image information at all outperforms some static saliency models[21,22]. Thus, scan path dynamics also play an important role. The ability of a model to predict human-like behavior can be much improved[16] by adding basic dynamic mechanisms to the static image-based predictions[15,21,23,24].

Theoretical[13,25] and experimental work[24] agree that two essential components in explaining dynamic scan paths are attentional selection and inhibitory tagging of previously fixated locations. The former refers to the combination of foveation and the attentional field, which defines a limited area from which information can be extracted. The attentional field is often represented as a Gaussian distribution, with its peak representing the fovea. Thus, as a first-order approximation, visual input is given by a Gaussian blob defined by the fixation position in a given scene. The second component keeps track of fixation history in order to drive exploration in scan paths and prevent continuous return to the same high-saliency regions[26]. In behavioral experiments, *inhibition of return* has been widely found as a component of human visual behavior[27], electrophysiology[28], and, more recently, as a neural process in the frontal eye field[29].

Attentional selection and inhibitory tagging have been previously implemented in a dynamical model for scan path generation[14,16]. The SceneWalk model[14] serves as a platform for the current work on the analysis of the role of attention around the time of saccade. Conceptually the model comprises two independent streams, activation and inhibition, which are computed on discrete $128 \times 128$ grids mapped to the image dimensions. The activation stream is implemented as a Gaussian

aperture around the current fixation location (see Eq. (1)) convolved with a saliency map. This local saliency then evolves over time using a differential equation (see "Methods" for mathematical details), meaning that past fixations can influence the current activation stream. The inhibition stream implements fixation tagging by Gaussian maps centered around the fixation location and similarly evolving over time using a differential equation such that past fixations retain some influence over the current inhibition stream. The size of the Gaussian window $\sigma_{A/F}$, as well as the decay parameters $\omega_{A/F}$ and other free model parameters are jointly obtained from the parameter inference (see "Methods"). As illustrated in Fig. 1, activation and inhibition maps are subtractively combined to yield a priority map[30], i.e., the 2D fixation probability map for the selection of the upcoming saccade target.

In the current context of perisaccadic processes, it is important to note that the strongest impact on mean fixation duration is generated by the variation in saccadic turning angles[15]. Continuing to move along the previous saccade's vector is associated with much shorter fixation durations than when the saccade direction changes by 90° or more (see the 80 ms effect in Fig. 6a). Therefore, we primarily seek to explain this coupling between fixation duration and saccade angle. Thus, we simplify our analysis by assuming random timing of fixation durations (assuming a gamma-distribution) and investigate the coupling with target selection under different turning angles. In future work the temporal control in the model could be extended to include other metrics (e.g., local saliency) for predicting fixation durations.

In this article we investigate a neurophysiologically plausible implementation of attentional dynamics and inhibitory principles. We extend the SceneWalk model[14] of eye-movement control by adding the concept of attentional shifts around the time of a saccade. Large-scale numerical simulations are carried out to estimate model parameters from experimental data using Bayesian data assimilation[16]. These covert perisaccadic attentional shifts turn out to improve model performance on a variety of eye-movement statistics.

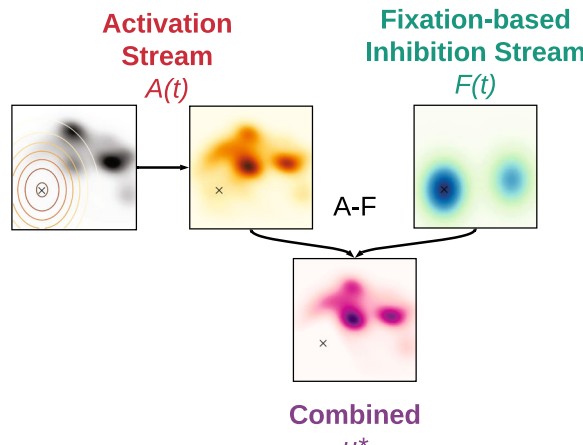

**Fig. 1 Attentional processing streams in a conceptual scan path model.** Visual attention and inhibitory tagging are largely independent processing streams which evolve neural activations via time-dependent input and decay. Constraining a saliency map (black and white color map) by a Gaussian aperture can approximate the extent of visual attention (orange color maps), as shown on the left. Inhibitory tagging, shown in blue color maps, keeps track of previously visited locations, as shown on the right. The 'x' marks the current fixation position. Combining the activation and inhibition streams yields a priority map from which fixation positions can be selected.

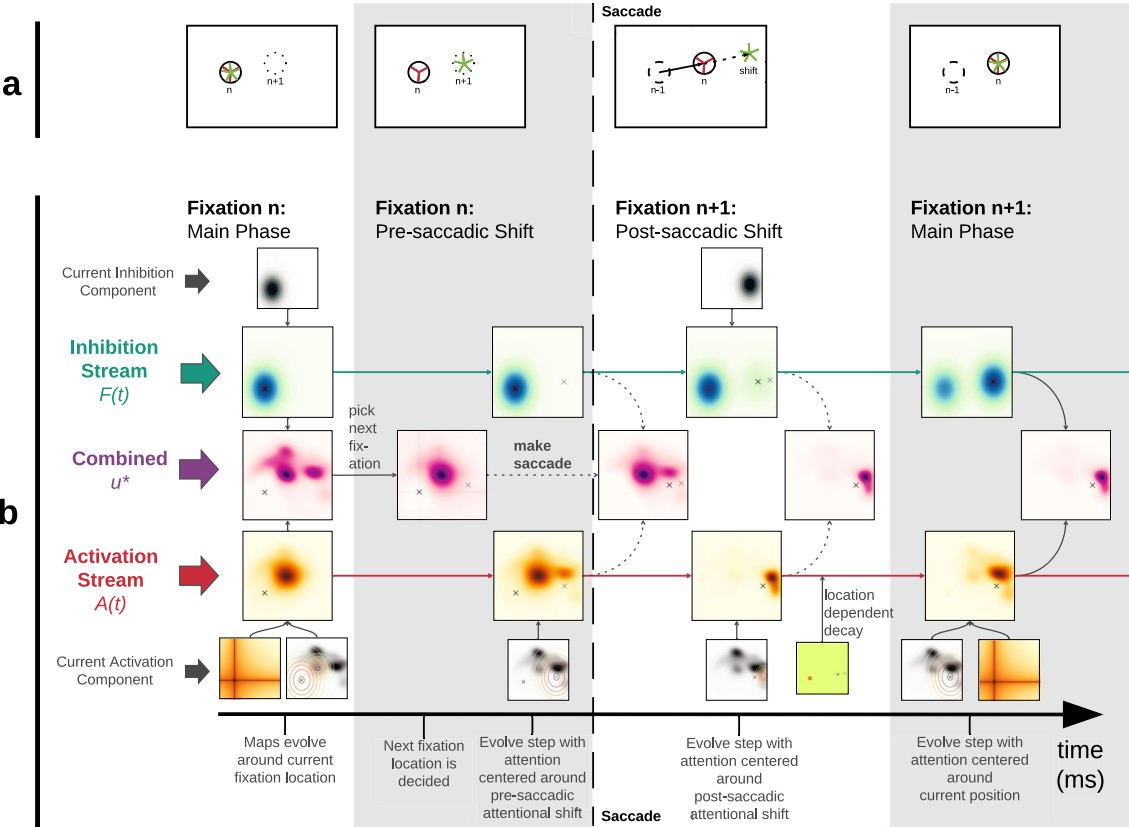

**Fig. 2 Timeline for processing around the time of saccade. a** Attention and fixation position. Leftmost panel: During fixations, locus of attention (green five-pointed star) and fixation position (red three-pointed star) are aligned. Second panel from left: Immediately before a saccade, the upcoming fixation location has already been selected; attention moves to the target location (green), while fixation position remains at launch site of the saccade. Third from left: After saccade execution, fixation position has been updated and, simultaneously, attention has shifted along the retinotopic activation trace (RAT) of the current fixation position before the saccade. Rightmost: During the fixation's main phase, the locus of attention and the fixation are realigned. **b** The activation (red-orange) and inhibition (blue-green) streams evolve over time during each of three model phases in each fixation. When a new fixation location is to be selected the streams are subtracted to yield a priority map (pink-purple). The activation map consists of a Gaussian aperture around a phase-dependent point in the image and image information as well as influences from past states of the model. The inhibition stream consists of a Gaussian aperture around the current fixation location and past states of the model.

## Results

The current work investigated the potential role of perisaccadic attention on human saccade statistics. In the next paragraph, we explain our theoretical model, before we describe experimental paradigm and experimental data.

**Integrating perisaccadic attention with gaze control.** Before the saccade is executed toward a target, performance benefits in accuracy and speed can be measured at the target location. This has frequently been interpreted as attention being allocated to the part of the image that is about to be fixated as part of saccadic planning. In Fig. 2a (leftmost), we see that during a fixation, the fixation location and the center of attention are coaligned. Once the upcoming target location is selected from the priority map $u_{ij}(t)$ but before the saccade occurs (Fig. 2a, second from left), attention already moves to the upcoming saccade target, decoupling fixation (red three-pointed star) and attention (green five-pointed star). The concept that covert attention shifts precede saccadic eye movements is well-established in the literature[10,31], with clear evidence for this *predictive attentional targeting* as early as 150 ms before saccade onset[32].

Furthermore, attention has been shown to move retinotopically with the saccade[33]. Thus, just after a saccade similar processing benefits can be found in a location along the saccade vector, which aligns with the retinotopic position of the target before the

saccade[34], a phenomenon called *retinotopic attentional trace* (RAT). The pre-allocated attention peak moves with the saccade such that it lands shifted along the saccade vector away from the saccade target. Figure 2a (third from left) shows that immediately after a saccade, attention is shifted to the same retinotopic position as the previous pre-saccadic shift and thus spatiotopically shifted in the same direction as the saccadic movement. Experimentally, the influence of the shift lasts about 100−200 ms[34]. After this interval the locus of activation moves to coincide with the fixation position again (Fig. 2a, rightmost panel). An alternative representation of the temporal progression of persaccadic processes in the model is available in Supplementary Fig. S2.

If we consider the added activation along the saccade vector as a component in saccade selection, this is in good agreement with the experimental finding of shorter fixation durations before forward saccades. The post-saccadic RAT is therefore the second part of the attentional decoupling that begins before saccade onset. Behavioral evidence for attentional shifts during a saccade[10] as well as neurophysiological correlates for post-saccadic retinotopic enhancements have been found[35]. Below we suggest that attentional shifts are a likely explanation for a systematic effect on saccade statistics observed during scan path formation. Figure 2b illustrates the influence of perisaccadic attentional shifts on the activation maps. The streams evolve over time (Eqs. (4) and (5)). Each successive map consists of the

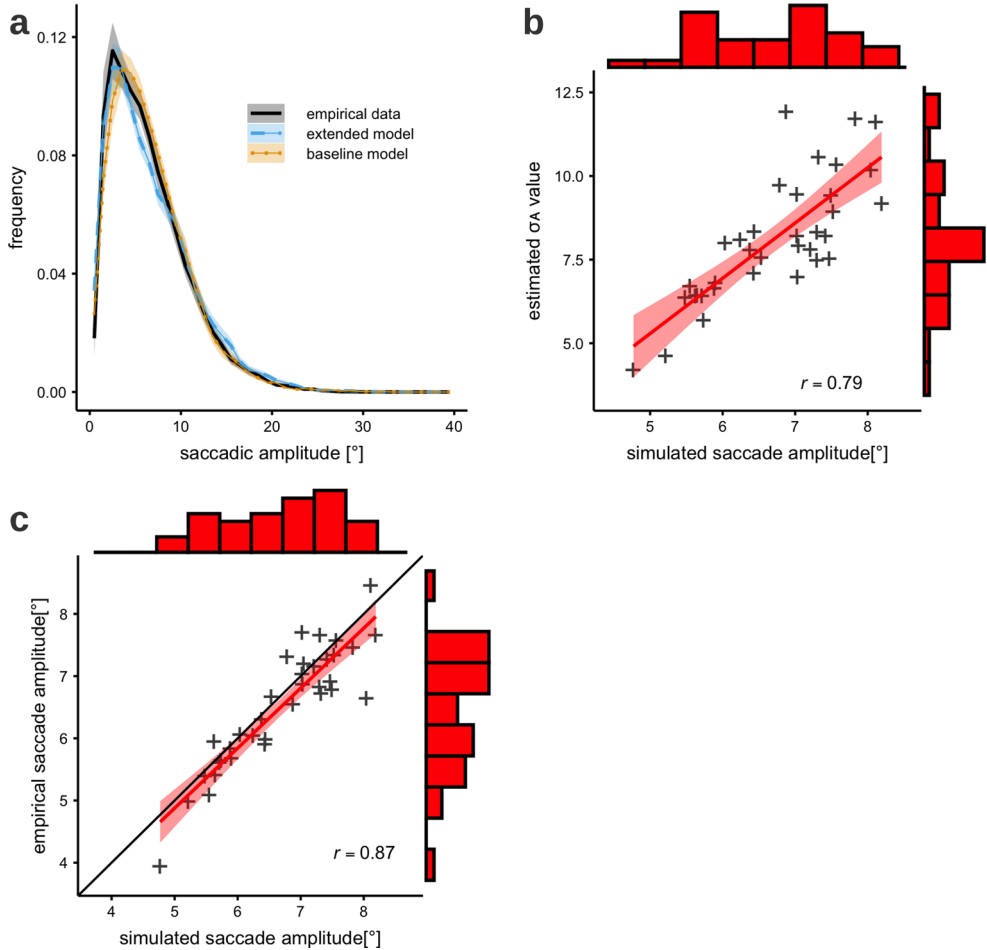

**Fig. 3 Saccade amplitude distribution and inter-individual differences. a** The saccade amplitude distribution for experimental data (black line), the baseline SceneWalk model (blue, dashed line) and the extended model (yellow, dotted line). Shading represents the 95% confidence interval between subjects. **b** Size parameter $\sigma_A$ of the attentional span and is positively correlated simulated mean saccade amplitude. **c** A high correlation is observed between experimental and simulated data on test images, using parameter estimates for each participant obtained for training data.

previous map and the current new information in a ratio determined by the decay function. The model, thus, has infinite memory, although depending on the strength of the decay parameters, previous fixation's influence may decrease rapidly Fixation targets are selected from the priority map (Eq. (6)) at time $t_{fix} - \tau_{pre}$, where $t_{fix}$ is the duration of the fixation and $\tau_{pre}$ is the duration of the pre-saccadic shift. Once the upcoming target is selected, attention moves to its location; after saccade execution, the post-saccadic attentional shift occurs; lastly, attention and fixation position are realigned when entering the main fixation phase (for details of the implementation, see Supplementary Information).

In the experiments, 35 human observers viewed 30 natural color images (see Supplementary Information). We will compare simulations for the *baseline model*[14,16] which includes only local saliency and inihibition evolving over time with the *extended model* that includes perisaccadic attention mechanisms. Model parameters for both models were estimated independently for each participant. For model fitting, fixation sequences of 2/3 of the images were used as training data, while all subsequent analyses were carried out on the remaining test images for each participant. The following section details some characteristic eye-movement statistics found in experimental data.

**Saccade amplitude distribution**. The distribution of saccade amplitudes generated during a scene viewing experiment varies

across participants and images. Overall, both the baseline and the extended models reproduce the qualitative shape of the saccade amplitude distribution (Fig. 3a)[36–39]. The experimentally observed saccade amplitude distribution is right-skewed, reflecting that amplitudes tend to be smaller than computer-generated saccades obtained by random sampling from the static 2D fixation density[6,14]. Previously, we suggested this drop in saccade amplitudes is caused by the foveated visual system, which preferentially selects saccade targets from within attentional span. Therefore, inter-individual differences in mean saccade amplitudes should correlate with the size of the attentional span $\sigma_A$, which is defined as the standard deviation parameter of the Gaussian-shaped attentional blob (see Eq. (1)). In Fig. 3b, we show the expected correlation between $\sigma_A$ and mean of saccade amplitude across participants, indicating that a larger area does indeed lead to longer saccades.

This statistic is perhaps the most prominent and intuitive. Previous modeling studies, like our baseline model, have been able to capture it as well as the extended model. The result we show here confirms that our addition of more complex mechanisms has not come at the cost of the more basic effects.

Additionally, the improved fitting procedure allows both models to be fit separately for each subject. With model parameters estimated for each participant using the training images, the predicted mean saccade amplitudes for test images were compared to experimentally observed mean saccade

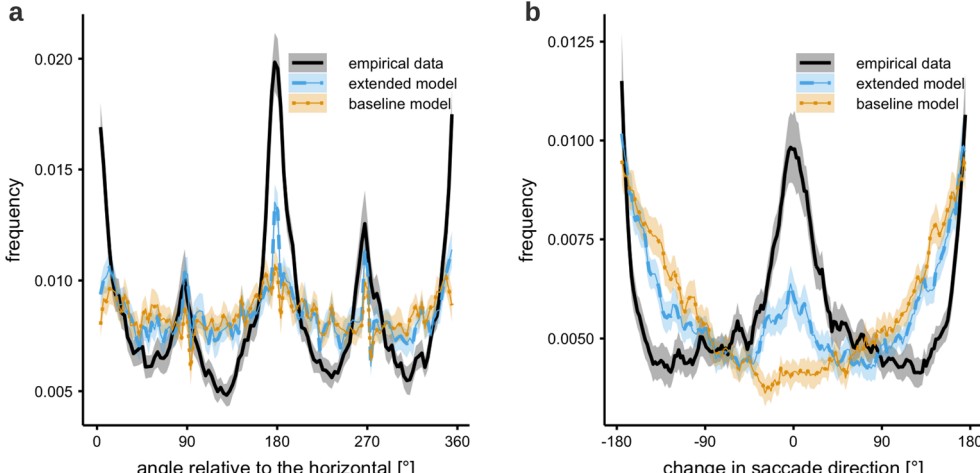

**Fig. 4 Saccade angle distributions. a** Absolute angle distribution in the empirical and simulated data. Empirical data show a strong tendency for saccading in the cardinal directions. This is strongly image dependent and not specifically considered in the models. The shading shows the 95% confidence interval between subjects. **b** Saccadic turning angle distribution in the empirical and simulated data. The angle shown is the divergence from the previous saccade direction. Empirically we find a tendency to continue along the previous saccade vector or completely reverse it. The extended model partly shows this behavior. The shading shows the 95% confidence interval between subjects.

amplitudes. We found good agreement between predicted and experimentally observed mean saccade amplitudes (Fig. 3c) indicated by a high correlation ($r = 0.91$). Our model is able to explain the inter-individual differences in the data via parameter variation.

**Absolute and relative saccade angle distributions**. Saccade angles are another important characteristic of human eye-movement behavior. The absolute angle distribution reports the directions of saccades relative to the image frame. Interestingly, there is a strongly image-dependent tendency, which varies mostly with the distribution of image features. On average the distribution shows characteristic peaks in the four cardinal directions[40,41]. Figure 4a shows that the baseline model does not show the pronounced pattern found in experimental data. Comparatively the extended model shows a clear improvement with distinct peaks at 0°, 90°, 180°, 270°, and 360°. The extended model implements a mechanism for an oculomotor potential (see Eqs. (14) and (15)), which preferentially weights the activation in the cardinal directions[42] before it is combined with the inhibition stream.

The saccade turning angle distribution characterizes the relationship of consecutive saccades. In the experimental data there is a clear bias towards forward saccades, which follow the same vector of motion, and a secondary preference for return saccades, which reverse the saccade vector. Therefore, we should expect clear peaks at 0° and 180° in the corresponding turning angle distribution[21,24,43,44]. Figure 4b shows the results of the baseline and extended models in comparison to experimental data. The baseline model produces a U-shaped distribution without any indication of a forward bias. There is an increased probability of turning by about 180°, since the edges of the image represent hard constraints. This effect is large enough to overshadow the effects of return saccades that directly return to the previous fixation location (of which there are comparatively few). The extended model does develop a peak for forward saccades, showing better qualitative agreement with the experimental data, although the bias towards forward saccades is clearly weaker than in the experiment. The model's slightly muted responses could be caused by a number of factors, not least of which is the fact that the chosen general purpose likelihood procedure does not specifically target this metric. The indirect

fitting of parameters supports the existence of the directional biases but may capture them only partially in the presence of other variance in the data.

The statistical preference of observers to maintain current saccade direction has been referred to as saccadic momentum[43–46]. Here we propose that the experimental effect is at least partially due to attentional enhancement in the current saccade direction, which generates a peak in the attention map that produced the forward bias.

**Joint probability of intersaccadic angle and amplitude**. More generally, we can identify potential dependencies of saccade turning angle and saccade amplitude by visualizing the corresponding joint probability (Fig. 5). As discussed above, compared to all other directions, there is a pronounced tendency for saccades to either maintain or completely reverse the direction of the previous saccade. This effect is well documented in the literature[21,24,38,43,44] and is independent of a variety of other factors such as image content. The values on the axes in Fig. 5 are relative to direction and amplitude of the previous saccade. In this normalized coordinate space, the previous saccade moved from position (−1, 0) to position (0, 0). The plotted density indicates the probability of the following saccade to be executed in a direction and with an amplitude relative to the previous saccade. Figure 5a reveals that there are two clear peaks in the experimental data, i.e., the return peak to the normalized launch site (−1, 0) of the previous saccade and the forward peak that is related to the saccadic momentum effect discussed above. It is important to note that the experimental return peak is not particularly high, but it is distinct since surrounding 2D regions do not exhibit a high fixation density.

In our extended model (Fig. 5b), the mechanism responsible for the forward saccades is the attentional shift before and after a saccade (Eqs. (9)−(11)). The distinctive shape of the return saccade peak, we suggest, is the result of the combination of a slow, global inhibition of return and a directed smaller facilitation of return (Eq. (12)) (see Supplementary Information). The former is implemented as the model's inhibition stream, while the latter is implemented as reduction in decay speed in the attention map, localized at the previous fixation location. The baseline model cannot produce the return and forward peak, since it lacks the mechanistic principles for coupling subsequent saccades.

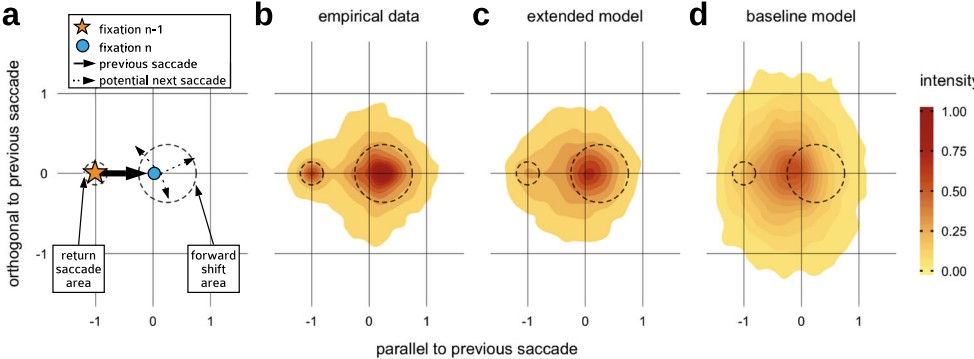

**Fig. 5 Joint probability of saccade turning angle and saccade amplitude normalized to the previous saccade. a** Legend for the joint probability plot. The coordinate system is normalized relative to the previous saccade. **b** The experimental probability shows the return and forward peaks. **c** The extended model captures these characteristic properties qualitatively. **d** In the baseline model, neither the return peak nor the forward peak can be found.

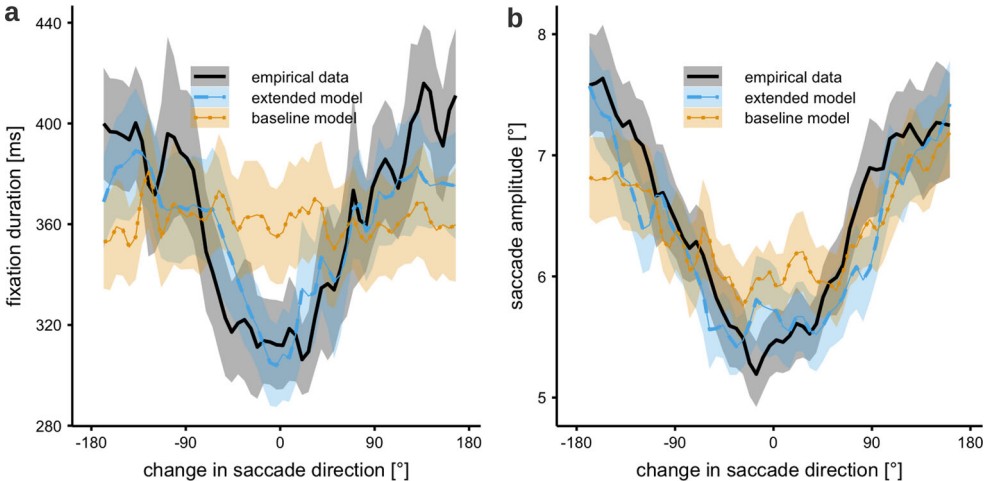

**Fig. 6 Average saccade amplitude and fixation duration are related to the change in saccade direction (saccadic turning angle). a** Fixation duration is shortest for saccade moving forward. Results from the extended model are in good agreement with the experimental data. **b** Saccade amplitude is smallest for forward saccades and largest for return saccades. While the baseline model reproduces this effect qualitatively, the extended model produces a better fit to the experimental data.

**Intersaccadic angle and fixation duration and saccadic amplitude**. The next two analyses correspond to the interdependence of fixation duration and saccade amplitude, and saccadic turning angles. Both have a distinctive shape in the data, showing that forward saccades tend to be shorter and preceded by shorter fixations, while changing direction takes longer and evokes longer saccades. Pilot simulations indicated that the effect reported in this section are not due to the addition of the oculomotor potential.

The new model notably improves the fit of the dependence of fixation duration on the turning angles (see Fig. 6). While previously there was no temporal component in the model, the added phases of shifted activation enable the model to dynamically respond to the duration of a fixation. In the model, each fixation begins with the post-saccadic shift phase. In terms of the attention activation map, this means that there is more activation along the previous saccade vector. After this phase the influence of the shift diminishes. Thus, when the fixation is short, there is still a lot of influence from the shift, increasing the chance of producing a forward saccade. When the fixation is long, the influence of the post-saccadic shift has subsided, allowing for activation from other salient locations to guide the saccade.

**Likelihood-based comparison**. Since our approach includes the likelihood computation of the baseline and extended models, we can make use of the models' likelihood functions for model comparison[16]. This approach entails evaluating the model likelihood given the empirical test data and computing the average log-likelihood per fixation of all scan paths. We then compare this metric to previous models[47].

The overall likelihood of the model given the data is larger for the extended model than for the original model (Fig. 7). In general, improved likelihood indicates improved predictive power of a model. The additions to the baseline model discussed in the current study, though theoretically well-founded, were extensive and considerably increased the model complexity. Conceivably adding these mechanisms could have led to improved scan path dynamics but worsened overall likelihood predictions, or else made the model volatile or unstable. In general, the likelihood is an objective measure of overall model performance[16]. As we have seen, the extended model performs much better than the baseline model at a number of qualitative eye-movement effects, while the improvement in general model likelihood is relatively small. Effects such as the impact of saccade turning angles on saccade amplitude are strong and important for biological plausibility of the model. At the same time, however, the impact on the overall likelihood is limited, since their contribution to 2D fixation density is small. In combination, the large improvements in eye-movement statistics and relative improvements in likelihood across model variants allow a strong conclusion in favor of the proposed model extension.

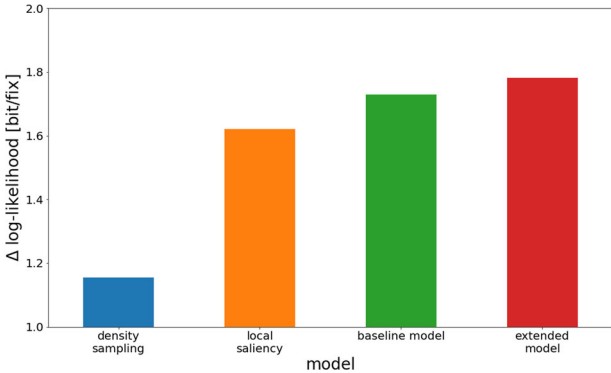

**Fig. 7 General model likelihood of models fitted on the training data given the test data.** Density sampling draws fixations directly from the empirical fixation distribution without dynamics. Local saliency produces scan paths by picking from the fixation density filtered through a Gaussian window, with no dynamics. The baseline model and the extended model are the dynamic models described in this article.

## Discussion

Moving from models of static fixation probabilities to the generation of scan paths has recently begun to attract interest in the field of attention modeling[14–16,23,48]. The success of saliency-based visual attention modeling[13,19,47] over the last 30 years makes a strong case for the use of priority maps[30] as a core component in scan path generation. In addition to image and task influences biologically represented in priority maps, scan paths on scenes are also characterized by a number of statistical characteristics, e.g., saccade angles and modulations of fixation duration or saccade amplitude by saccadic turning angles. Our modeling study lends support to the fact that attentional dynamics around the time of saccade exert a fundamental influence on the behavioral statistics of scan paths.

Previous research on visual attention shows that processing resources are covertly allocated away from the current fixation location just before[10,31,32] and just after[32,34,35] a saccade is produced. In this study, we added shifts of covert attention to a dynamical model of scan path generation[14,16] and find improved agreement with gaze statistics observed in experimental data. Most importantly, the characteristic distribution of saccadic turning angles with a clear bias towards forward and return saccades and the influences of saccadic turning angle on fixation durations and saccade amplitudes can be explained by covert attention shifts around the time of a saccade. The importance of covert attention and perisaccadic mechanisms is apparent throughout the visual system, both at the macroscopic and at the microsaccade levels[49–51].

The first generation of computational models in scene viewing were static models that predicted fixation locations on any given image based on statistical image features. The strength of these static models lies in producing densities that resemble empirical fixation density maps. Recently, the predictive power of some models has become close to perfect and approached the gold standard[19,47]. However, by design these models do not take temporal dynamics within a scan path and the inhomogeneity of the retinal acuity into account. From this perspective, it is not surprising that static models predict fixation density, but not sequences of fixations[16,24,52]. This simple fact points to the interesting observation that eye movements in scene viewing are guided in large part, but not exclusively by observer- and image-specific factors. Human eye movements are influenced by oculomotor and attention systems, producing pervasive systematic statistical tendencies in experimental data.

Previously published dynamic models outperform static models substantially[16,23]. The most evident feature of the human visual system that indisputably influences scan path dynamics is foveation. Accordingly, even a minimal model like weighting a saliency map by the distance to a current fixation location significantly improves model performance[53]. The SceneWalk model[14], which served as a baseline for our study, incorporates foveated saliency in its activation stream. A further advance in the modeling of scan paths has been the addition of inhibitory fixation tagging[26,54,55]. The baseline model implements such an inhibition stream as a second component shaping the priority map[30] by difference of activation.

The fact that long fixations often occur in frequently fixated areas[56] implies that fixation duration and target selection are related. The LATEST model[15] combines the prediction of scan paths and fixation durations by interpreting scan paths as a continuous series of stay (maintain fixation) or go (saccade) decision[57–59]. Each individual location on a weighted saliency map influences two LATER units[60], i.e., one for normal and long latencies and one for short latencies. These units accumulate evidence from each location in the image until one reaches a threshold depending on the current location, triggering a saccade. The accumulation rate of each location in the image is controlled by image-content factors like image features and semantic interest, as well as by oculomotor factors like the change in saccade direction and target eccentricity. Coupling of experimental data and model is achieved by statistical linear mixed-effects modeling. Thus, the LATEST model makes little attempt at explaining the origin of the factors that influence the rate of evidence accumulation, instead focusing on the specific selection mechanism and its relationship with fixation duration. By contrast, the extended SceneWalk model is based on mechanistic assumptions derived from neural and cognitive knowledge about the contributing factors to fixation selection. Parameters are based on statistically rigorous likelihood approach that evaluates the model assumptions given the data.

Generally, the value of a model must be quantified in terms of predictive power and explanatory value. For the models discussed here, we carried out comparisons of simulated scan paths and human eye-movement data. A number of metrics have been proposed for such a comparison[61–63]. Critically, however, the choice of individual statistics has a crucial influence on the outcome and there is, in most cases, no rigorous justification for the used metric. A solution to this is to evaluate dynamical scan path models using a likelihood approach[16], which provides a statistically well-founded and reliable measure for the predictive quality of a dynamical model. In this article we relied on Bayesian data assimilation[64] as a statistically rigorous framework for testing whether the model architecture accurately represents the data generation process. This approach turned out to be particularly fruitful for strongly theory-guided models. Using general likelihood to estimate parameters of the model lends credibility to the theoretical foundations from eye-movement literature implemented by the model.

In addition to better predicting human scan paths during scenes viewing, the integration of biologically inspired attentional dynamics into models of eye guidance unifies two very disparate fields of eye-movement research. The research into covert attention shifts and perisaccadic effects is typically concerned with processes that occur on a highly detailed level in very controlled experimental setups. By contrast scene viewing literature usually operates at a higher level, on which the minutia of saccade programming or covert attention are typically passed over. Thus, influences arising from the microscopic level of eye-movement control can explain effects we observe at the macroscopic level.

## Methods

**Experiment.** Experimental data for this study were collected in a larger corpus study on scene viewing which is described in detail elsewhere[65,66]. Images and fixation data from this corpus experiment can be downloaded from an Open Science Foundation repository (see below[66]). The corpus consists of eye-movement data from 105 participants viewing 90 images of natural or urban landscapes from six different categories for a fixed duration (10 s). Each category contained 15 images. Images were chosen such that the most interesting image parts either fell on the left, right, upper, lower or, central image side (Supplementary Fig. S1 provides some examples). The last category were images with natural patterns, minimizing the influence of particularly salient objects. During the viewing subjects were given no task except to freely view the images.

In this study we used Experiment 3 from the corpus study[65,66], in which participants viewed color images. This subset of data contains the eye movements of 35 participants, who viewed 30 images from each category without a task. We further split the data set into test and training data by randomly choosing 1/3 of the images (ten from each category) for each participant.

For saccade detection we applied a velocity-based algorithm[67,68]. Saccades had a minimum amplitude of 0.5° and exceeded an average velocity during a trial by six (median-based) standard deviations for at least six data samples (12 ms). The epoch between two subsequent saccades was defined as a fixation. After preparation, 312,267 fixations and saccades were detected for further analysis.

**Baseline model.** The original SceneWalk model[25] was implemented on a $128 \times 128$ grid, where $(x, y)$ give the physical coordinates in degrees. For each fixation in the scan path we start by computing simple 2D Gaussians centered at current fixation position $(x_f, y_f)$ for both the inhibition and the attention pathway, each with an appropriate standard deviation $\sigma_{A/F}$ (A denotes the attention stream, F denotes the fixation stream to generate inhibitory tagging).

$$G_{A/F}(x,y) = \frac{1}{2\pi\sigma_{A/F}^2}\exp\left(-\frac{(x-x_f)^2 + (y-y_f)^2}{2\sigma_{A/F}^2}\right). \quad (1)$$

Both the inhibition $F_{ij}(t)$ and the activation $A_{ij}(t)$ streams evolve over time under current visual input and decay (due to limited of visual memory), i.e.,

$$\frac{dA_{ij}(t)}{dt} = \omega_A\left(\frac{S_{ij}\,G_A(x_i, y_j; x_f, y_f)}{\sum_{kl}S_{kl}\,G_A(x_k, y_l; x_f, y_f)} - A_{ij}(t)\right), \quad (2)$$

$$\frac{dF_{ij}(t)}{dt} = \omega_F\left(\frac{G_F(x_i, y_j; x_f, y_f)}{\sum_{kl}G_F(x_k, y_l; x_f, y_f)} - F_{ij}(t)\right), \quad (3)$$

where the input to the activation maps is the Gaussian-weighted local saliency $S_{kl}G_A(x_k, y_l; x_f, y_f)$ and the input to the inhibition map is a Gaussian blob at current fixation position.

The differential equations that determine the temporal evolution of the activation maps, Eq. (2) for the attention map and Eq. (3) for the fixation/ inhibition map, can be integrated analytically to provide a closed solution for the activation changes during fixation, i.e.,

$$A(t) = \frac{G_A S}{\sum G_A S} + e^{-\omega_A(t-t_0)}\left(A_0 - \frac{G_A S}{\sum G_A S}\right), \quad (4)$$

and

$$F(t) = \frac{G_F}{\sum G_F} + e^{-\omega_F(t-t_0)}\left(F_0 - \frac{G_F}{\sum G_F}\right), \quad (5)$$

where we dropped the indices $i$, $j$ for simplicity. In the equations, the term $e^{-\omega_{A/F}(t-t_0)}$ determines the speed of decay of the past states of the map.

Next, both activation maps were combined to compute the priority map $u_{ij}(t)$,

$$u_{ij}(t) = \frac{\left(A_{ij}(t)\right)^\gamma}{\sum_{kl}\left(A_{kl}(t)\right)^\gamma} - C_F\frac{\left(F_{ij}(t)\right)^\gamma}{\sum_{kl}\left(F_{kl}(t)\right)^\gamma}. \quad (6)$$

Mathematically, the two maps are shaped by exponent $\gamma$ before subtraction, and a weight parameter $C_F$ for inhibition is introduced. We expect $\gamma \approx 1$, equivalent to Luce's choice rule[69].

As subtraction can cause negative activation, in the next step we take only the positive component of the map,

$$u_{ij}^*(u_{ij}) = \begin{cases} u_{ij} & \text{if } u_{ij} > 0 \\ 0 & \text{otherwise} \end{cases} \quad (7)$$

and, finally, add noise $\zeta$

$$\pi(i,j) = (1-\zeta)\frac{u_{ij}^*}{\sum_{kl}u_{kl}^*} + \zeta\frac{1}{\sum_{kl}1} \quad (8)$$

to obtain the probability map $\pi(i, j)$ for the selection of saccade targets. This process is repeated for each fixation in a sequence, where the current state information is combined with the past activation maps to produce a continuously evolving prediction of the next fixation.

**Table 1 Model phases: onset times and locations around which the Gaussians in both streams are centered.**

| Phase | Start | End | F center | A center |
|---|---|---|---|---|
| Post-saccadic shift | 0 | $\tau_{\text{post}}$ | $\text{fix}_n$ | remap |
| Main (no shift) | $\tau_{\text{post}}$ | $t_{\text{fix}} - \tau_{\text{pre}}$ | $\text{fix}_n$ | $\text{fix}_n$ |
| Pre-saccadic shift | $t_{\text{fix}} - \tau_{\text{pre}}$ | $t_{\text{fix}}$ | $\text{fix}_n$ | $\text{fix}_{n+1}$ |

Parameter $t_{\text{fix}}$ indicates the fixation's duration, parameters $\tau_{\text{pre}}$, $\tau_{\text{post}}$ are the phase durations, and parameters $\text{fix}_{n+1}$, $\text{fix}_n$, and remap are the locations.

The model structure reveals the following parameters: (1, 2) $\sigma_A$ and $\sigma_F$, which are the standard deviations of the current fixation's attention and inhibition Gaussians respectively, (3, 4) $\omega_A$ and $\omega_F$, which are the speed at which past states of the model lose influence over the current, (5) $\gamma$, the shaping parameter for the Gaussians, (6) the coupling factor $C_F$, which is the weight of the inhibition pathway, and (7) the noise parameter $\zeta$ determining the background noise for the probability map $\pi(i, j)$.

**Pre-saccadic attentional shifts.** Once a new fixation location is chosen the center of attention moves to the upcoming fixation location, while the center of the inhibition map remains at the current fixation location (see Table 1). In the model, the pre-saccadic shift is implemented by moving the attentional Gaussian to center around the next fixation location, while the inhibition remains in the same position for a time $\tau_{\text{pre}}$. The inhibition stream is calculated for the entire fixation duration using Eq. (5), therefore, we have

$$G_A^{\text{pre}}(x,y) = \frac{1}{2\pi\sigma_A^2}\exp\left(-\frac{(x-x_{f+1})^2 + (y-y_{f+1})^2}{2\sigma_A^2}\right), \quad (9)$$

and then continue computations using Eqs. (4) and (5) with $G_A^{\text{pre}}$ instead of $G_A$ for the duration of $\tau_{\text{pre}}$. When the pre-saccadic phase terminates, the saccade is executed.

**Post-saccadic attentional shifts.** The center of the post-saccade attention peak is determined by extending the vector of the preceding saccade by a shift amplitude $\eta$, i.e.,

$$(x_s, y_s) = (x_n, y_n) + \frac{(x_\delta, y_\delta)}{\sqrt{x_\delta^2 + y_\delta^2}} \cdot \eta, \quad (10)$$

where the saccade direction is given by the vector $(x_\delta, y_\delta)$ with $x_\delta = x_n - x_{n-1}$ and $y_\delta = y_n - y_{n-1}$. Thus, the attentional Gaussian is centered at the shifted location

$$G_A^{\text{post}}(x,y) = \frac{1}{2\pi\sigma_{\text{post}}^2}\exp\left(-\frac{(x-x_s)^2 + (y-y_s)^2}{2\sigma_{\text{post}}^2}\right) \quad (11)$$

during the post-saccadic phase. Temporal evolution of activation maps continues based on Eqs. (4) and (5) with $G_A^{\text{post}}$ instead of $G_A$ for a duration of $\tau_{\text{post}}$. Meanwhile, the inhibition stream evolves with the center of inhibition in the same location as the fixation position.

After the post-saccadic shift phase, the cycle is completed and another main phase follows. The attention center moves to each of the three locations in turn via discrete steps as shown in Table 1. We have chosen this discrete approximation with constant durations of pre- and post-saccadic shifts to compute activation changes in all fixation phase efficiently. Neurophysiological support for our discrete approximation has been found[35], indicating that attention does not move smoothly over space from location $n$ to location $n + 1$ but instead selectively starts building up at the target location $n + 1$.

**Facilitation of Return.** To account for Facilitation of Return (FoR), we implement a selectively slower decay of the attention map in a spatial window centered at the previous fixation location. Different from the overall decay rate $\omega_A$, we define a reduced decay rate $\omega_{\text{FoR}}$ for a window $x - v < x_{f-1} < x + v$ and $y - v < y_{f-1} < y + v$ around the previous fixation location $(x_{f-1}, y_{f-1})$, where $v$ is the size of the window. Therefore, reduced decay of activation in the attention map, Eq. (4), is given by

$$A(t) = \frac{G_A S}{\sum G_A S} + e^{-\omega_{\text{FoR}}(t-t_0)}\left(A_0 - \frac{G_A S}{\sum G_A S}\right) \quad (12)$$

for the spatial window defined above.

In addition to the strongly attention-related mechanisms above, we added the following two less dynamic and more general biases.

**Center bias.** The original SceneWalk model initiates its activation maps with uniform distributions. While it is difficult to accurately know the initial state of the visual system when viewing images, previous work has shown that the central

**Table 2 Maximum posterior density estimates of the model parameter estimations of all subjects and credibility intervals (see text).**

| Parameter | Baseline SW mean | Baseline SW +/− | Extended SW mean | Extended SW +/− |
|---|---|---|---|---|
| $\omega_A$ | 14.802 | 2.555 | 9.996 | 2.391 |
| $\sigma_A$ | 7.482 | 1.165 | 7.320 | 1.004 |
| $\sigma_F$ | 4.629 | 1.041 | 6.834 | 2.626 |
| $\gamma$ | 0.935 | 0.095 | 0.956 | 0.102 |
| $\log(\zeta)$ | −1.132 | 0.131 | −1.727 | 0.260 |
| $\chi$ | — | — | 0.059 | 0.028 |
| $\eta$ | — | — | 0.415 | 0.105 |
| $\log(\psi)$ | — | — | −0.613 | 0.192 |

fixation bias has a strong influence on the first fixation. Starting the model with a central activation improves the predictions of the model[70]. In line with this finding, we also initiated the model with central activation. The evolution equation for the first fixation is

$$A(t) = \frac{G_{\text{fix}}S}{\sum G_{\text{fix}}S} + e^{-\omega_{\text{CB}}(t-t_0)}\left(A_{0_{\text{CB}}} - \frac{G_{\text{fix}}S}{\sum G_{\text{fix}}S}\right). \quad (13)$$

**Oculomotor potential**. Research into the oculomotor system has revealed a marked preference for saccades in the cardinal directions. In order to implement this tendency in the model, we introduced an additive oculomotor component. A plus-shaped oculomotor map centered on the current fixation position is generated

$$\text{OMP} = \left((x - x_f)^2 \cdot (y - y_f)^2\right)^\chi, \quad (14)$$

where the factor $\chi$ determines the steepness of the slopes. The oculomotor map is added to the combined map $u_{ij}$, before the normalization and the addition of noise (Eqs. (7) and (8))

$$u_{\text{OMP}} = u + \left(\psi \cdot \left|-\frac{\text{OMP}}{\sum(\text{OMP})}\right|\right). \quad (15)$$

**Additional model parameters**. The implementation of the extended SceneWalk model gives rise to several new parameters. To the seven parameters of the original model, we add (a) $\omega_{\text{CB}}$, the decay speed of the center bias; (b, c) $\sigma_{\text{CB}_x}$ and $\sigma_{\text{CB}_y}$, the size of the center bias; (d, e) $\tau_{\text{pre}}$ and $\tau_{\text{post}}$, the durations of the attention shift phases; (f) $\eta$, the distance of the post-saccadic shift; (g) $\sigma_{\text{post}}$ the size of the shifted Gaussian; (h, i) $\omega_{\text{FoR}}$, the attention decay at the previous fixation position and $\nu$, the size of the facilitation window; and (j, k) the steepness $\chi$ and factor $\psi$ of the oculomotor potential.

**Estimated and fixed model parameters**. We implemented a fully Bayesian approach to parameter inference[16] using numerical computation of the models' likelihood functions and advanced Monte Chain Monte Carlo (MCMC) techniques. Details are given in the next section. For a discussion of the full results including marginal posterior densities, see Supplementary Note 4: Detailed results on parameter estimation.

In Table 2 we report point estimates for all parameters as averages over participants. The full estimates for each participant can be found in the Supplementary Information (Tables S2 and S3). These point estimates were computed from the posterior densities by determining the highest posterior density region for an alpha of 0.5 (i.e., the highest 50% of the density are in this region), assuming a unimodal distribution. The reported credibility intervals the lower and upper bounds of the highest density interval. The point estimate for the parameter represents the center of the highest posterior density interval.

Some of the model parameters could be constrained by the physiological literature and some of the parameters had to be fixed in order to improve convergence of the parameter estimation. The latter case was checked by large-scale pilot simulations with different model versions using a separate data set. In Table S1 we list all fixed model parameters.

First, we separated the time scales of attention and inhibition stream by one order of magnitude, i.e., $\omega_F = \omega_A/10$. We assume $\omega_F$ is slower to decay by a magnitude than $\omega_A$, to enable long-term inhibition of return and fast build-up of activation for attentional capture. Second, we set $C_F = 0.3$, where the numerical value was obtained from pilot simulations indicating that the relative influence of the inhibition stream must be smaller (but not negligible) compared to the corresponding influence of the attention stream.

In the extended model, some of the additional parameters need further discussion. First, we set $\sigma_{\text{CB}} = 4.3$ and $\omega_{\text{CB}} = 1.5$ as described in ref. [70], for a typically sized center bias and an attention decay that is slower than the regular $\omega$.

The center bias parameters are difficult to estimate, since their influence is mainly limited to the first fixation. Second, we fixed $\omega_{\text{FoR}} = \omega_A/10$, representing an approximate value for decay slower by a magnitude and the size of the facilitation of return window to be approximately the size of the fovea, i.e., 2° of visual angle. As before, only a relatively small amount of fixations are influenced by this mechanism, making it difficult to identify the numerical value reliably. Third, we set the times post- and pre-saccadic attentional shifts to $\tau_{\text{pre}} = 0.1$ s and $\tau_{\text{post}} = 0.05$ s, where the numerical values are determined by pilot simulations. Due to their small magnitude, values for $\zeta$ and $\psi$ were estimated in the log scale.

**Bayesian parameter inference**. Parameter inference of the dynamical models discussed here was implemented in the general framework of data assimilation[64] using a fully Bayesian estimation procedure[16,71,72]. In this statistical inference we used the computation of the models' likelihood functions. Given a fixation sequence $f_1 \ldots f_{i-1}$, where each fixation $f_i$ is determined by its coordinates $f_i = (x_i, y_i)$, the likelihood of the model specified by a set of parameters $\theta$ can be computed as a product of probabilities, i.e.,

$$L_M(\theta|\text{data}) = P_M(f_1) \cdot \prod_{i=2}^{n} P_M(f_i|f_1, \ldots, f_{i-1}, \theta), \quad (16)$$

where $P_M(f_1)$ is the probability of the initial fixation starting at time $t = 0$ and the conditional probabilities $P_M(f_i|f_1 \ldots f_{i-1}, \theta)$ can be read off from the models priority map $\pi(i, j)$.

For scaling and numerical reasons the log-likelihood is usually used. Thus, the sum of the scan path's log-likelihood per fixation for the entire data set gives one value that characterizes model performance. As suggested by ref. [16], taking the $\log_2$ of the likelihood enables the use of the unit bit. A null model, in which the probability of choosing each point a $128 \times 128$ pixel image is the constant, would be $\log_2(1/128^2) = -14$. A hypothetical model which, unrealistically, perfectly predicts the data would have a log-likelihood of 0. It is important to note that for model comparison we can take the mean log-likelihood per fixation while for the parameter estimation the non-normalized sum log-likelihood of a scan path is the appropriate measure.

Based on the likelihood $L_M(\theta|\text{data})$ and a prior distribution $P(\theta)$, the posterior distribution is computed via Bayes' rule as

$$P(\theta|\text{data}) = \frac{L(\theta|\text{data})P(\theta)}{\int_\Omega P(\theta)L(\theta|\text{data})d\theta}, \quad (17)$$

where typically a Markov Chain Monte Carlo (MCMC) approach is needed to compute the posterior density numerically. For our parameter estimations we used the implementation of the DREAM Algorithm that is published as PyDream[73]. Each estimation ran three chains of 20,000 iterations. Since the DREAM estimation procedure requires a large number of model evaluations, the computing time of the likelihood function is critical for the baseline SceneWalk model and, in particular, for the extended SceneWalk model. We therefore implemented parallel computations of the likelihood for fixation sequences. The priors, loosely based on pilot estimations on a separate data set, were chosen to be broad and relatively uninformative.

Inter-individual differences in behavior are a main source of variance in eye-movement data. Here we took advantage of these differences by testing model generalizability. We implemented individual independent model fitting for each participant by running a DREAM parameter estimation for each participant separately. The advantage of using this method is that when simulating data, we obtain an upper limit for the variance of parameters between individual participants.

**Reporting summary**. Further information on research design is available in the Nature Research Reporting Summary linked to this article.

## Data availability
The experimental data used in this study represent a subset of the Potsdam Corpus on Spatial Frequency Search in Natural Scenes[66], which is publicly available via the Open Science Framework (osf.io/caqt2).

## Code availability
Source code used for numerical simulations, analyses, and plotting as well as other project-related files are made available (osf.io/qsx4w).

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

## Acknowledgements

We thank Noa Malem-Shinitski, Maximilian Rabe, Stefan A. Seelig, and Silvia Makowski for valuable discussions. This work was supported by a grant from Deutsche Forschungsgemeinschaft, Germany (SFB 1294, project no. 318763901). We acknowledge a grant for computing time from Norddeutscher Verbund für Hoch- und Höchstleistungsrechnen (HLRN), Germany (grant no. bbx00001).

## Author contributions

L.S., L.O.M.R., R.E., and H.A.T. designed the research and analyzed the experiment. L.S., L.O.M.R., and R.E. developed the model, carried out the simulations, and analyzed the results. L.S. and R.E. wrote the manuscript. L.S., L.O.M.R., R.E., and H.A.T. reviewed the manuscript.

## Funding

## Competing interests

The authors declare no competing interests.
