## [Peer Review File · Communications Biology]

Reviewers' comments:

Reviewer #1 (Remarks to the Author):

In this paper, authors propose a new computational model for predicting the visual scanpaths of an observer. For that, and this is the main novelty, the visual scanpaths are estimated thanks to perisaccadic covert attention. To my knowledge, this is the first attempt of this kind in this field. Overall, I really enjoyed reading the paper. The proposed model is, from my point of view, a framework leveraging and simulating perisaccadic attention mechanisms. I really like and this is the key point of this paper. The performances are improved compared to the baseline model. That's good.

The weakness of the paper is the validation and the proof that the perisaccadic attention mechanisms really help (conceptually this is very nice to embed such mechanisms in the modelling). The evaluation of such model is an open-issue, that is mentioned by authors in the discussion.

Please find below my comments:

(i) In the abstract, the link or transition between the second sentence and the following sentence is not clear for me. Authors mix computational modelling of visual attention with intrinsic properties of gaze deployment. Could authors please rewrite the very first sentences of the abstract?

(ii) In introduction, what do authors mean by "... they are normally aligned under natural viewing conditions"? I get the message by reading the following sentence but perhaps the transition could be improved?

(iii) Authors wrote that "the most advanced models are not far from predicting perfect fixation density maps". I would really tone down this claim. They perform rather well in a very constrained environment when the ecological validity of the experiment is more than questionable (e.g. screen, no head movement, static color images, universal saliency maps (see my comment below regarding the population))... In the same paragraph, authors wrote that bottom-up image information is crucial for prediction saliency maps. Crucial is perhaps too strong, since when we use an image-blind saccadic model, this one perform very well compared to handcrafted saliency model. See the following references:

Tatler, B. W., & Vincent, B. T. (2009). The prominence of behavioural biases in eye guidance. *Visual Cognition*, 17(6-7), 1029-1054.

Le Meur, O., & Coutrot, A. (2016). Introducing context-dependent and spatially-variant viewing biases in saccadic models. *Vision research*, 121, 72-84.

(iv) Below formula (2), authors wrote that the input to the activation maps is the Gaussian-weighted local saliency. How do authors estimate the local saliency? By the way, the caption of Figure 1 should be improved and explained the meaning of the left-hand side image as well as the cross (which is obviously a fixation but...). According to figure 1, it seems that the fixation history is limited only to one fixation (see the greenish map called fixation-based inhibition stream). Could authors make this point clear?

(v) The last paragraph of page 2 is very hard to read. I would suggest to re-write it a little bit to make easier the reading. Note as well that I do not understand why reference 33 is associated with fixation duration and saccade angle?

(vi) The section titled Integrating perisaccadic attention with gaze control is very interesting. Fig 2 illustrates very well the proposed method. Note that on Fig 2 I see the fixation history much better than in Fig 1. That's good, but it could be improved in my opinion.

(vii) In the Method, the pre-saccadic and post-saccadic attention shifts are well explained. Even if I understand the computational description, I have some difficulties to understand how they affect the results. However, I really believe that the proposed method brings added values in the computational prediction of scanpaths.

For instance, in Fig. 3A, authors plot the saccade amplitude distribution for baseline model and extended model. As the saccade amplitude is the Euclidean distance between fixation points, I do not get how the perisaccadic attention mechanisms make the thing better on this point. But, on this point, I do not see the reasons why results are better... Could you please elaborate on this

point? Is it due to the model parameters that have been estimated for each participant using the training images? If yes, is it fair to compare with the baseline model (if not trained in a similar way)?

Regarding the absolute and relative saccade angle distributions, the extended model performs better since it "implements a mechanism for an oculomotor potential". It would be nice to refer to Eq. 14 and 15. So for this part, I understand the benefit of the extended model over baseline one. Regarding Joint probability of intersaccadic angle and amplitude, the benefit of the extended model is clear. However, I cannot say that this is due to perisaccadic attention mechanisms. It could be due to the oculomotor potential authors rightly introduced.

In addition, the axis represented the amplitude of the previous saccade is not that clear for me. It would be better to express this axis in visual degree, no?

Something that authors could discuss is the huge discrepancy within observers, difference between children and adults, between neurotypical and ASD people, and so on. May I suggest the following articles:

Le Meur, O., Coutrot, A., Liu, Z., Rämä, P., Le Roch, A., & Helo, A. (2017). Visual attention saccadic models learn to emulate gaze patterns from childhood to adulthood. *IEEE Transactions on Image Processing*, 26(10), 4777-4789.

Helo, A., Pannasch, S., Sirri, L., & Rämä, P. (2014). The maturation of eye movement behavior: Scene viewing characteristics in children and adults. *Vision research*, 103, 83-91.

Typos:

Page 10: "of of the inhibition

Reference 22: "LeMeur" should be in two words. 9

Best, Olivier

Reviewer #2 (Remarks to the Author):

In this modeling study, Schwetlick et al., examine the link between peri-saccadic attention and gaze statistics during scene viewing. They report that their models reproduce, to varying degrees, various gaze related statistics such as saccade amplitude distribution, angular statistics, inter-saccadic turning angles, and so on.

This is an important topic of study. However, the manuscript as written is very dense and difficult to read. It fails to bring out the novelty of the research topic and the impact of the results.

My major comments are:

(a) The introduction is very sparse and not well motivated. Some concepts are introduced without any preamble whatsoever. For example the author state: "... it is important to note that the strongest impact on fixation duration is generated by the variation in saccadic turning angles". Why is this important?

Why is this important?

(b) There are repeated references to a "baseline model" and an "extended model" throughout the text. It is very hard to tell the differences between the two models without digging into the references and the methods section. An adequate background should be provided for the "baseline" model without expecting the reader to look up other references. Further, the differences between the baseline model and the extended model should be clearly enunciated before diving into the Results.

(c) I do not understand the logic of stating Equations 1-3 early on in the manuscript, while the components of these equations appear in Equations 4-6 in the Methods. I would suggest relegating all equations to Methods (so that the Methods section is coherent) and give the readers an

intuitive description of the models in the main text.

(d) The authors state: "Figure 4A shows that the baseline model does not show the pronounced pattern found in experimental data. Comparatively the extended model shows a clear improvement with distinct peaks at 0°, 90°, 180°, 270°, and 360°." Visually, the blue and yellow traces in Fig 4A are very similar. What is the basis of the authors' assertion?

(e) In the section titled Likelihood-based comparison, the authors state: "As we have seen, the extended model performs much better than the baseline model at a considerable number of eye-movement statistics, while the improvement in general model likelihood is quite small. In combination, the large improvements in eye-movement statistics and relative improvements in likelihood across model variants allow a strong conclusion in favor of the proposed model extension". I do not understand the logic behind this.

I would recommend that the authors rewrite the manuscript for clarity.

Please also check for grammatical errors throughout the text.

Reviewer #3 (Remarks to the Author):

This is a very nice paper. It was a real pleasure to read. It combines the previous model of the authors on predicting static fixation densities with new components modelling peri-saccadic phenomena, in order to provide a better estimate of dynamic scan-path behavior during the viewing of natural scenes.

The paper is well written and clear (except for some comments on Figs. 1 and 2 below). And, the model is plausible.

I have no major comments.

One general idea that came to my mind while reading this paper is that it reminded me of Tian et al., Front. Sys. Neurosci. 2016 in the realm of microsaccades and Posner cueing. In that case, with the task being one of fixating a tiny fixation spot, the scan path of the microsaccades is really of the fixation spot. However, the authors tried to link their other evidence that peri-microsaccadic "attentional" phenomena can spread in space away from the microsaccade endpoints (e.g. Chen et al., Curr Biol 2015 and Hafed, 2013). So, they asked what would happen if one goes as far as modelling the entire behavior (in such simplified Posner tasks and not natural images) as arising **solely** through peri-microsaccadic phenomena (and not other processes like peripheral covert attention). The model worked really well in capturing both attentional capture and inhibition of return (without a need to know a priori where the previous cue was to dictate behavioral performance; performance was solely determined by peri-microsaccadic phenomena). And, simple parameter changes also accounted for individual differences quite well. This suggests that peri-movement phenomena are sufficient to account for that simple behavior. Of course, this was in only a simplified Posner task, not with rich natural stimuli. And, so, in the current study, it makes sense to consider everything (like the bottom-up image driven saliency maps etc). However, I wonder if the authors could comment on linking these two fields, because it does seem to me that they are linked by the idea that peri-saccadic changes in the visual system are really important to consider for explaining behavior.

Some minor comments below:

- Fig. 1 is not very clear. I had a hard time interpreting it. For example, what is the x and what are the contours in the top left? And, what are the different color coded maps? The grey? The colors? etc. I think it would help if the legend was more clear. Also, the legend doesn't make it clear whether this is a general figure or a figure describing the specific hypothesis in the paper? I know

that these things get mentioned in the text, but just browsing through the figures, there was an uncertainty in my mind about what this figure was saying: is this state of the art or the current hypothesis? It would be great if the authors expanded the legend to better explain the figure.

- top of p. 3, for the sentence "predictive attention targeting as early as 150 ms before saccade onset": I have a bit of an issue with this sentence. It is hard to justify a link between an event 150 ms before a saccade and the saccade itself. Presenting probes at different times from saccade onset (the usual way of assessing attention) might work for peri-saccadic probe times of $\sim <50$ ms or so. However, once you go longer, the probe interacts with the whole system and can even cancel saccades before reprogramming (e.g. saccadic inhibition). So, whatever emerges from such very early probes relative to saccade onset (in the case of 150 ms) is likely a result of multiple brain processes and not just a predictive allocation of attention. 150 ms is a whole lot of time indeed, and a lot of things can happen in the brain.

- very last paragraph of introduction: the text up to here was really nice and beautiful to read. Then, looking at Fig. 2B, things got quite difficult. The figure seems to be very difficult to read. I guess one issue is that it is hard to visualise which of the activity maps shift with time and which do not. For example, what is the relationship between the final little blob to the right of fixation in the priority map (dark magenta blob with a lighter vertical line of activation below it) to the priority map at the very beginning? I wonder if one could perhaps add a one-dimensional panel simplifying the ideas by describing a 1-dimensional version of the different activity shifts etc. This would help to at least solidify the concepts, especially given that we are still in the introduction section at this stage.

- Fig. 4: can the authors comment on why the model is a bit muted in its biases in relation to the empirical data?

- fig. 6: it's interesting that in the realm of microsaccades where the scan path is really at the fixation spot, there seems to be consistent evidence. e.g. Tian et al., Journal Neurophysiology, 2018.

Revision of manuscript COMMSBIO-20-1321 “Modeling the effects of perisaccadic attention on gaze statistics during scene viewing” by Schwetlick et al.

Detailed responses to reviewer’s comments:

Reviewer #1 (Remarks to the Author):

In this paper, authors propose a new computational model for predicting the visual scanpaths of an observer. For that, and this is the main novelty, the visual scanpaths are estimated thanks to perisaccadic covert attention. To my knowledge, this is the first attempt of this kind in this field.

Overall, I really enjoyed reading the paper. The proposed model is, from my point of view, a framework leveraging and simulating perisaccadic attention mechanisms. I really like and this is the key point of this paper. The performances are improved compared to the baseline model. That’s good.

The weakness of the paper is the validation and the proof that the perisaccadic attention mechanisms really help (conceptually this is very nice to embed such mechanisms in the modelling). The evaluation of such model is an open-issue, that is mentioned by authors in the discussion.

Answer: We thank the reviewer for the positive overall evaluation of our study. In the revision, we tried to improve the description of our model validation.

Please find below my comments:

(i) In the abstract, the link or transition between the second sentence and the following sentence is not clear for me. Authors mix computational modelling of visual attention with intrinsic properties of gaze deployment. Could authors please rewrite the very first sentences of the abstract?

A: We fully agree with the reviewer’s comment and changed the third sentence of the abstract accordingly.

(ii) In introduction, what do authors mean by “... they are normally aligned under natural viewing conditions”? I get the message by reading the following sentence but perhaps the transition could be improved?

A: We thank the reviewer for this comment and changed the corresponding sentence in the introduction (line 18).

(iii) Authors wrote that “the most advanced models are not far from predicting perfect fixation density maps”. I would really tone down this claim. They perform rather well in a very constrained environment when the ecological validity of the experiment is more than questionable (e.g. screen, no head movement, static color images, universal saliency maps (see my comment below regarding the population))... In the same paragraph, authors wrote that bottom-up image information is crucial for prediction saliency maps. Crucial is perhaps too strong, since when we use an image-blind saccadic model, this one perform very well compared to handcrafted saliency model. See the following references:

Tatler, B. W., & Vincent, B. T. (2009). The prominence of behavioural biases in eye guidance. *Visual Cognition*, 17(6-7), 1029-1054.

Le Meur, O., & Coutrot, A. (2016). Introducing context-dependent and spatially-variant viewing biases in saccadic models. *Vision research*, 121, 72-84.

A: We agree with the reviewer that we might have overstated the predictive power of saliency models. Consequently, we toned down our statement. We thank the reviewer for pointing out the two relevant references which we cite in the revised version (line 29, 31).

(iv) Below formula (2), authors wrote that the input to the activation maps is the Gaussian-weighted local saliency. How do authors estimate the local saliency? By the way, the caption of Figure 1 should be improved and explained the meaning of the left-hand side image as well as the cross (which is obviously a fixation but...). According to figure 1, it seems that the fixation history is limited only to one fixation (see the greenish map called fixation-based inhibition stream). Could authors make this point clear?

A: We thank the reviewer for raising this point, which we think we now describe more clearly. The size of the attention window is part of the estimation procedure, the corresponding size parameter σ_A is a free model parameter that is estimated numerically from data (line 51). We also moved the full model equations, incl. Eqs. (1) to (3), to the Methods (line 259) and hope that the set of all model parameters is more clearly indicated in the revised manuscript.

(v) The last paragraph of page 2 is very hard to read. I would suggest to re-write it a little bit to make easier the reading. Note as well that I do not understand why reference 33 is associated with fixation duration and saccade angle?

A: We thank the reviewer for this suggestion and re-wrote the paragraph. We also removed the reference from this paragraph (line 54 and following).

(vi) The section titled Integrating perisaccadic attention with gaze control is very interesting. Fig 2 illustrates very well the proposed method. Note that on Fig 2 I see the fixation history much better than in Fig 1. That's good, but it could be improved in my opinion.

A: We agree that our description could be improved here. Therefore. We revised Fig. 1 and the corresponding figure captions. We also revised the description of Fig. 2 in the text.

(vii) In the Method, the pre-saccadic and post-saccadic attention shifts are well explained. Even if I understand the computational description, I have some difficulties to understand how they affect the results. However, I really believe that the proposed method brings added values in the computational prediction of scanpaths.

For instance, in Fig. 3A, authors plot the saccade amplitude distribution for baseline model and extended model. As the saccade amplitude is the Euclidean distance between fixation points, I do not get how the perisaccadic attention mechanisms make the thing better on this point. But, on this point, I do not see the reasons why results are better... Could you please elaborate on this point?

A: We agree with the reviewer that our explanation is incomplete here. Since we are extending an existing model, I must be shown that the extended, more complex model can still explain the basic statistics (saccade amplitude distribution). Therefore, we added the following sentences to the description of the saccade amplitude plots: "This statistic is perhaps the most prominent and intuitive. Previous modeling studies, like our baseline model, have been able to capture it as well as the extended model. The result we show here confirms that our addition of more complex mechanisms has not come at the cost of the more basic effects." (line 113 and following).

We also added the corresponding references to the Methods when discussing specific effects and the proposed mechanisms in the model (e.g., in the paragraph on “Joint probability of intersaccadic angle and amplitude) (line 156).

Is it due to the model parameters that have been estimated for each participant using the training images? If yes, is it fair to compare with the baseline model (if not trained in a similar way)?

A: We added a statement on this issue, making more transparent that parameters for baseline and extended models were estimated independently for each participant’s experimental data. The bottomline is that we apply exactly the same procedure to baseline and extended models.(line 99, 116)

Regarding the absolute and relative saccade angle distributions, the extended model performs better since it “implements a mechanism for an oculomotor potential”. It would be nice to refer to Eq. 14 and 15. So for this part, I understand the benefit of the extended model over baseline one.

A: We thank the reviewer for the comment and added the reference to the oculomotor equations (line 127).

Regarding Joint probability of intersaccadic angle and amplitude, the benefit of the extended model is clear. However, I cannot say that this is due to perisaccadic attention mechanisms. It could be due to the oculomotor potential authors rightly introduced.

A: We agree with the reviewer that the specific effects of the additional mechanism in the extended model are not discussed. In pilot simulations, it turned out that the intersaccadic angle and amplitude effects were not created by the addition of the oculomotor potential. We added a statement to the corresponding section in the Results (line164).

In addition, the axis represents the amplitude of the previous saccade is not that clear for me. It would be better to express this axis in visual degree, no?

A: We agree that the figure might have been difficult to interpret. We now revised Figure 5 and added a new A panel showing how the saccade amplitude was normalized. In a normalized coordinate space, the previous saccade moved from position $(-1,0)$ to position $(0,0)$. The plotted density indicates the probability of the following saccade to be executed in a direction and with a length relative to the previous saccade (lien 149).

Something that authors could discuss is the huge discrepancy within observers, difference between children and adults, between neurotypical and ASD people, and so on. May I suggest the following articles:

Le Meur, O., Coutrot, A., Liu, Z., Rämä, P., Le Roch, A., & Helo, A. (2017). Visual attention saccadic models learn to emulate gaze patterns from childhood to adulthood. *IEEE Transactions on Image Processing*, 26(10), 4777-4789.

Helo, A., Pannasch, S., Sirri, L., & Rämä, P. (2014). The maturation of eye movement behavior: Scene viewing characteristics in children and adults. *Vision research*, 103, 83-91.

A: We added the suggested reference to the Appendix where we discuss interindividual difference in detail (line 559).

Typos:

Page 10: “of of the inhibition

Reference 22: “LeMeur” should be in two words.

A: Thank you, corrected.

Reviewer #2 (Remarks to the Author):

In this modeling study, Schwetlick et al., examine the link between peri-saccadic attention and gaze statistics during scene viewing. They report that their models reproduce, to varying degrees, various gaze related statistics such as saccade amplitude distribution, angular statistics, inter-saccadic turning angles, and so on.

This is an important topic of study. However, the manuscript as written is very dense and difficult to read. It fails to bring out the novelty of the research topic and the impact of the results.

A: We thank the reviewer for the positive evaluation of our research. In the revision, we tried to improve the presentation of our results to explain the novelty and relevance of our work.

My major comments are:

(a) The introduction is very sparse and not well motivated. Some concepts are introduced without any preamble whatsoever. For example the author state: “ ... it is important to note that the strongest impact on fixation duration is generated by the variation in saccadic turning angles ”. Why is this important?

A: We re-wrote the corresponding paragraph (also suggested by reviewer #1), explaining that the observed effect in mean fixation duration under variation of the saccade turning angle (Fig. 6A) is important because of its size of 80 ms (a fourth to a third of the mean fixation duration). We also refer to Tatler et al.'s (2017) model where it is shown that saccade angle are most important for predicting fixation durations (line 54 and following).

(b) There are repeated references to a “baseline model” and an “extended model” throughout the text. It is very hard to tell the differences between the two models without digging into the references and the methods section. An adequate background should be provided for the “baseline” model without expecting the reader to look up other references. Further, the differences between the baseline model and the extended model should be clearly enunciated before diving into the Results.

A: We thank the reviewer for raising this point. In the introductory/theoretical part of our manuscript, we now explain in more detail how baseline and extended model relate to each other (eg. line 43, 61, 99).

(c) I do not understand the logic of stating Equations 1-3 early on in the manuscript, while the components of these equations appear in Equations 4-6 in the Methods. I would suggest relegating all equations to Methods (so that the Methods section is coherent) and give the readers an intuitive description of the models in the main text.

A: We agree with the reviewer and moved all equations to the Methods in the revised version of the manuscript (line 259 and following).

(d) The authors state: "Figure 4A shows that the baseline model does not show the pronounced pattern found in experimental data. Comparatively the extended model shows a clear improvement with distinct peaks at 0°, 90°, 180°, 270°, and 360°." Visually, the blue and yellow traces in Fig 4A are very similar. What is the basis of the authors' assertion?

A: We fitted the models using the object likelihood-based approach (see next point). To re-check our model's performance we ran new simulations with one additional free parameter relating to the oculomotor potential (χ , see Eq. (14 and 15)). These simulations confirmed that for all of the four peaks (0, 90, 180, 270 degrees) the extended model performed better (peak height more similar to experimental data) than the baseline model.

(e) In the section titled Likelihood-based comparison, the authors state: "As we have seen, the extended model performs much better than the baseline model at a considerable number of eye-movement statistics, while the improvement in general model likelihood is quite small. In combination, the large improvements in eye-movement statistics and relative improvements in likelihood across model variants allow a strong conclusion in favor of the proposed model extension". I do not understand the logic behind this.

A: We thank the reviewer for raising this point and clarified our discussion in the revised manuscript. Improving models must always be reflected in a higher likelihood. This is the case in the current study. However, the improvement can be quite small (the baseline model already was a useful model for the data). The reason is that the extended model reproduces some smaller effects that are important for biological plausibility, while they produce an apparently small improvement in the overall likelihood. We rewrote this point in the revised manuscript (line 185).

I would recommend that the authors rewrite the manuscript for clarity.

Please also check for grammatical errors throughout the text.

A: We rewrote portions of the manuscript and carefully checked grammatical errors.

Reviewer #3 (Remarks to the Author):

This is a very nice paper. It was a real pleasure to read. It combines the previous model of the authors on predicting static fixation densities with new components modelling peri-saccadic phenomena, in order to provide a better estimate of dynamic scan-path behavior during the viewing of natural scenes.

The paper is well written and clear (except for some comments on Figs. 1 and 2 below). And, the model is plausible.

I have no major comments.

A: We thank the reviewer for this positive overall evaluation of our manuscript.

One general idea that came to my mind while reading this paper is that it reminded me of Tian et al., *Front. Sys. Neurosci.* 2016 in the realm of microsaccades and Posner cueing. In that case,

with the task being one of fixating a tiny fixation spot, the scan path of the microsaccades is really of the fixation spot. However, the authors tried to link their other evidence that peri-microsaccadic “attentional” phenomena can spread in space away from the microsaccade endpoints (e.g. Chen et al., *Curr Biol* 2015 and Hafed, 2013). So, they asked what would happen if one goes as far as modelling the entire behavior (in such simplified Posner tasks and not natural images) as arising *solely* through peri-microsaccadic phenomena (and not other processes like peripheral covert attention). The model worked really well in capturing both attentional capture and inhibition of return (without a need to know a priori where the previous cue was to dictate behavioral performance; performance was solely determined by peri-microsaccadic phenomena). And, simple parameter changes also accounted for individual differences quite well. This suggests that peri-movement phenomena are sufficient to account for that simple behavior. Of course, this was in only a simplified Posner task, not with rich natural stimuli. And, so, in the current study, it makes sense to consider everything (like the bottom-up image driven saliency maps etc). However, I wonder if the authors could comment on linking these two fields, because it does seem to me that they are linked by the idea that peri-saccadic changes in the visual system are really important to consider for explaining behavior.

A: We thank the reviewer for sharing these thoughts with us. Indeed, we believe that peri-saccadic phenomena should be qualitatively the same in microsaccades and “normal” saccades. We added a remark to the Discussion citing the work of Tian et al. (2016) (line 201).

Some minor comments below:

- Fig. 1 is not very clear. I had a hard time interpreting it. For example, what is the x and what are the contours in the top left? And, what are the different color coded maps? The grey? The colors? etc. I think it would help if the legend was more clear. Also, the legend doesn't make it clear whether this is a general figure or a figure describing the specific hypothesis in the paper? I know that these things get mentioned in the text, but just browsing through the figures, there was an uncertainty in my mind about what this figure was saying: is this state of the art or the current hypothesis? It would be great if the authors expanded the legend to better explain the figure.

A: We thank the reviewer for this comment and added an improved figure description for Figure 1.

- top of p. 3, for the sentence “predictive attention targeting as early as 150 ms before saccade onset”: I have a bit of an issue with this sentence. It is hard to justify a link between an event 150 ms before a saccade and the saccade itself. Presenting probes at different times from saccade onset (the usual way of assessing attention) might work for peri-saccadic probe times of ~ <50 ms or so. However, once you go longer, the probe interacts with the whole system and can even cancel saccades before reprogramming (e.g. saccadic inhibition). So, whatever emerges from such very early probes relative to saccade onset (in the case of 150 ms) is likely a result of multiple brain processes and not just a predictive allocation of attention. 150 ms is a whole lot of time indeed, and a lot of things can happen in the brain.

A: We agree with the reviewer that 150 ms is a long time interval when relating to brain processes. However, it should be noted that we “switch on” a differential equation 150 ms from saccade onset. This does not mean that the biological activations are instantaneously behaviorally relevant at this time. The build-up of activation in the maps just starts and will need a time on the order of 100 ms to control behavior. Thus, further simulation could be done in future work to relate our time scale to neurophysiological evidence.

- very last paragraph of introduction: the text up to here was really nice and beautiful to read. Then, looking at Fig. 2B, things got quite difficult. The figure seems to be very difficult to read. I guess one issue is that it is hard to visualise which of the activity maps shift with time and which do not. For example, what is the relationship between the final little blob to the right of fixation in the priority map (dark magenta blob with a lighter vertical line of activation below it) to the priority map at the very beginning?

A: We thank the reviewer for issue this point and tried to clarify this point in the text.

Also changed the description in the text: “Each successive map consists of the previous map and the current new information in a ratio determined by the decay function. The model, thus, has infinite memory, although depending on the strength of the decay parameters, previous fixation’s influence may decrease rapidly.” (line 88).

I wonder if one could perhaps add a one-dimensional panel simplifying the ideas by describing a 1-dimensional version of the different activity shifts etc. This would help to at least solidify the concepts, especially given that we are still in the introduction section at this stage.

A: We thank the reviewer for this suggestion. We created the figure below to this purpose and decided to include the figure in the Appendix (not in the main text, since Fig. 2A contains the same 1D information effectively). (line 516, referenced in line 79)

- Fig. 4: can the authors comment on why the model is a bit muted in its biases in relation to the empirical data?

A: We added the following comment to the text: “The model's slightly muted responses could be caused by a number of factors, not least of which is the fact that the chosen general purpose likelihood procedure does not specifically target this metric. The indirect fitting of parameters supports the existence of the directional biases but may capture them only partially in the presence of other variance in the data.” (line 137 and 184).

- fig. 6: it’s interesting that in the realm of microsaccades where the scan path is really at the fixation spot, there seems to be consistent evidence. e.g. Tian et al., Journal Neurophysiology, 2018.

A: We added a citation to the work by Tian et al. (2018) in the Discussion to point out that there is consistent evidence in the realm of microsaccades (line 201).

EOF

REVIEWERS' COMMENTS:

Reviewer #1 (Remarks to the Author):

Thanks a lot for all complementary information.

My first feeling was very positive on this scientific contribution. The revised draft has been really improved.

So I am happy to recommend this paper for publication.

Reviewer #2 (Remarks to the Author):

The authors have adequately responded to my previous comments in their revised manuscript. The updated manuscript is much clearer. I have no further comments other than a couple of very minor points:

- 1) On line 55 there is a "?" at the end of the sentence "... saccade target angles", instead of a paper reference
- 2) Lne 62: "mode"  "model"

Reviewer #3 (Remarks to the Author):

The authors have addressed all previous comments. Just some minor typos:

- line 55: question mark instead of a citation
- In Fig. 2B, the term "Phase Main" is hard to understand. Shouldn't it be "Main Phase"?
- line 104: "experiments" should be "experiment"

Revision of manuscript COMMSBIO-20-1321 “Modeling the effects of perisaccadic attention on gaze statistics during scene viewing” by Schwetlick et al.

Author Responses

Reviewer #1: Thanks a lot for all complementary information.

My first feeling was very positive on this scientific contribution. The revised draft has been really improved.

So I am happy to recommend this paper for publication.

Response: We thank the reviewer for the positive feedback and helpful comments during the review process.

Reviewer #2: The authors have adequately responded to my previous comments in their revised manuscript. The updated manuscript is much clearer. I have no further comments other than a couple of very minor points:

1) On line 55 there is a "?" at the end of the sentence "... saccade target angles", instead of a paper reference

2) Lne 62: "mode"  "model"

Response: We thank the reviewer for the positive feedback and constructive remarks. All typos have been corrected.

Reviewer #3: The authors have addressed all previous comments. Just some minor typos:

- line 55: question mark instead of a citation

- In Fig. 2B, the term “Phase Main” is hard to understand. Shouldn’t it be “Main Phase”?

- line 104: “experiments” should be “experiment”

Response: We thank the reviewer for the positive evaluation of our research and helpful comments. All typos have been corrected.